

# Application of artificially intelligent systems for the identification of discrete fossiliferous levels

David M. Martín-Perea[1,2,3], Lloyd A. Courtenay[4], M. Soledad Domingo[5] and Jorge Morales[1]

[1] Palaeobiology Department, Museo Nacional de Ciencias Naturales - CSIC, Madrid, Spain
[2] Geodynamics, Stratigraphy and Palaeontology Department, Universidad Complutense de Madrid, Madrid, Spain
[3] Institute of Evolution in Africa, Madrid, Spain
[4] Department of Cartographic and Land Engineering, Higher Polytechnic School of Avila, University of Salamanca, Avila, Spain
[5] Sciences, Social Sciences and Mathematics Department, Universidad Complutense de Madrid, Madrid, Spain

Corresponding author
David M. Martín-Perea,
davidmam@ucm.es

## ABSTRACT

The separation of discrete fossiliferous levels within an archaeological or paleontological site with no clear stratigraphic horizons has historically been carried out using qualitative approaches, relying on two-dimensional transversal and longitudinal projection planes. Analyses of this type, however, can often be conditioned by subjectivity based on the perspective of the analyst. This study presents a novel use of Machine Learning algorithms for pattern recognition techniques in the automated separation and identification of fossiliferous levels. This approach can be divided into three main steps including: (1) unsupervised Machine Learning for density based clustering (2) expert-in-the-loop Collaborative Intelligence Learning for the integration of geological data followed by (3) supervised learning for the final fine-tuning of fossiliferous level models. For evaluation of these techniques, this method was tested in two Late Miocene sites of the Batallones Butte paleontological complex (Madrid, Spain). Here we show Machine Learning analyses to be a valuable tool for the processing of spatial data in an efficient and quantitative manner, successfully identifying the presence of discrete fossiliferous levels in both Batallones-3 and Batallones-10. Three discrete fossiliferous levels have been identified in Batallones-3, whereas another three have been differentiated in Batallones-10.

## INTRODUCTION

The Batallones Butte, located 30 km to the south of Madrid (Spain, Fig. 1A) and 1 km to the east of Valdemoro (Fig. 1B), is home to nine Late Miocene paleontological sites (named Batallones-1, Batallones-2 and so on). These sites hypothetically correspond to hourglass-shaped cavities with upper openings (Fig. 1C), formed as a consequence of pseudokarstic processes (*Pozo et al., 2004*; *Calvo et al., 2013*), where mammals became
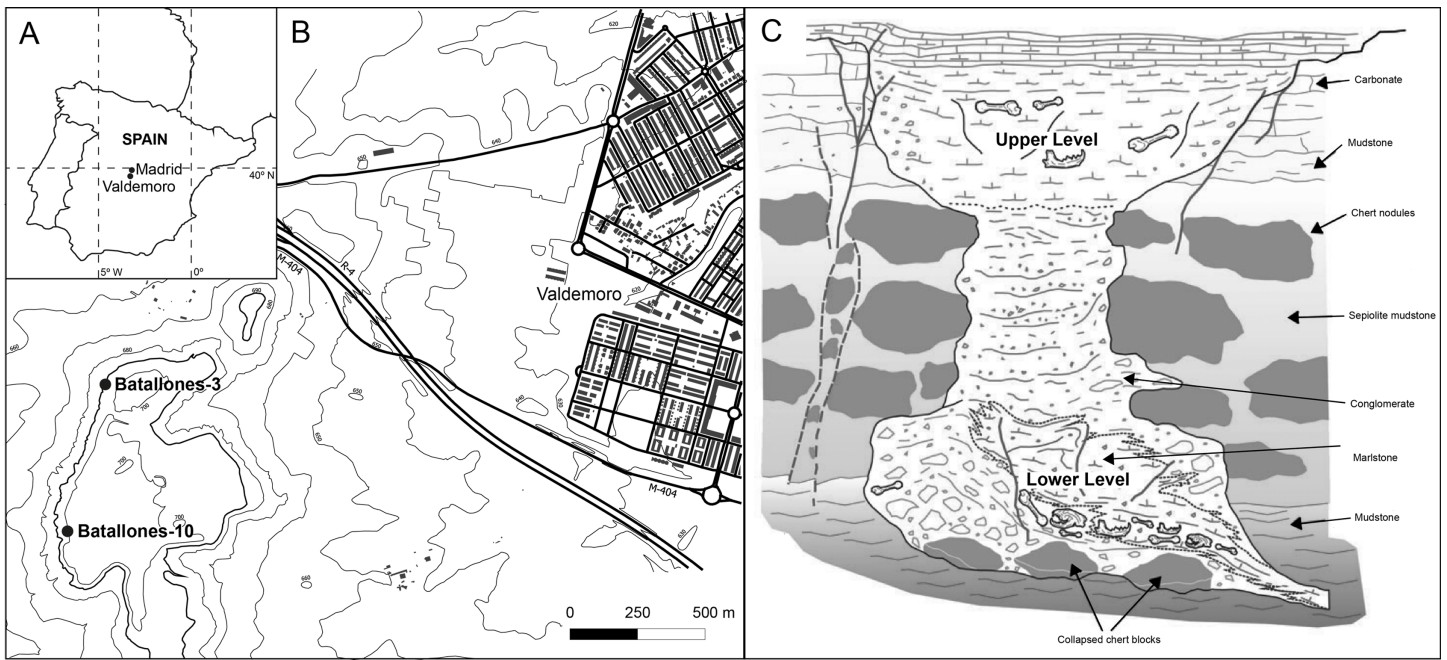

**Figure 1 Geographical and geological background.** (A) Map of Spain showing the location of the cities of Madrid and Valdemoro. (B) Detailed map of the situation of the Batallones Butte sites Batallones-3 and Batallones-10. (C) Hypothetical geomorphology and geology of Batallones Butte cavities, with an herbivore-dominated upper part and a carnivore-dominated lower cavity (modified from *Calvo et al. (2013)*).

trapped (*Domingo et al., 2011*, *2012*, *2013a*). The upper part of this hour-glass structure is formed by deposits where mammalian herbivore bones predominate, whereas the lower part of the structure is overwhelmingly dominated by carnivoran remains (more than 90% of the fossils). Both assemblages contain abundant, diverse and well-preserved remains (*Domingo et al., 2013a*). These cavities were located in a landscape composed by woodland with patches of wooded grassland (*Domingo et al., 2013b*, *2016*)

The Batallones Butte sites have a late Vallesian age (ca. 9.1 Ma; early Late Miocene), based on the faunal association (*Morales et al., 1992*, *2004*; *Domingo, Alberdi & Azanza, 2007*; *Morales et al., 2008*; *López-Antoñanzas et al., 2010*; *Gómez Cano, Hernández Fernández & Álvarez-Sierra, 2011*). However, micromammal studies have shown that site formation was not synchronic between sites, with Batallones-10 being older than Batallones-1, which in turn is older than Batallones-3 (*López-Antoñanzas et al., 2010*).

The decimeter to centimeter-scale separation of discrete fossiliferous levels within the same lithostratigraphic unit is common in Cenozoic sites (*Canals, Vallverdú & Carbonell, 2003*; *Uribelarrea et al., 2014*; *Sañudo, Blasco & Fernández Peris, 2016*; *Gravina et al., 2018*; *Martín-Perea et al., 2019*), but tend to be qualitative, using two-dimensional transversal and longitudinal projection planes made in continuous strips (*Canals, Vallverdú & Carbonell, 2003*). Using similar methods, *Martín Escorza & Morales (2005)* preliminarily inferred discrete fossiliferous levels at Batallones-1 using spatial data from the first 3 years of the excavation (2,273 remains, 17.19% of the total Batallones-1 excavated sample).

Palaeoecological, palaeoenvironmental and taphonomical studies carried out at Batallones Butte (*Antón & Morales, 2000*; *Domingo et al., 2011*, *2013a*, *2013b*, *2016*) have treated the upper and lower part of the hourglass-shaped cavities separately based on their different depth and their different taxonomical composition (herbivores vs. carnivores, respectively). Due to the challenges posed by the structure of the deposits that filled the Batallones Butte cavities, such as lateral facies changes, deformations, depositional asymmetries, collapse structures, slickensides and/or local tilting typical of cave deposits (*Calvo et al., 2013*), geological sub-levels have not been identified within each of these parts. However, the question still remains on whether or not discrete fossiliferous levels are found embedded in the apparently geologically continuous deposits.

The pioneering introduction of Artificially Intelligent Algorithms (AIAs) in fields of archaeology and paleoanthropology has revolutionized numerous sub-disciplines such as those related with genetic sequencing (*Mondal, Bertranpetit & Lao, 2019*), site and object detection (*Anemone, Emerson & Conroy, 2011*; *Conroy et al., 2012*; *Emerson & Anemone, 2012*; *Emerson et al., 2015*; *Benhabiles & Tabia, 2016*; *Block et al., 2016*; *Wills, Choiniere & Barrett, 2018*; *Anemone & Conroy, 2018*; *Caspari & Crespo, 2019*; *Verschoof-van der Vaart & Lambers, 2019*), physical anthropology (*Bewes et al., 2019*), biomechanics (*Püschel et al., 2018*) restoration (*Derech, Tal & Shimshoni, 2018*; *Hermoza & Sipiarn, 2018*), as well as taphonomy (*Arriaza & Domínguez-Rodrigo, 2016*; *Domínguez-Rodrigo, 2019*; *Egeland et al., 2018*; *Byeon et al., 2019*; *Courtenay et al., 2019*; *Moclán, Domínguez-Rodrigo & Yravedra, 2019*).

AIAs, including those trained through Machine Learning and Deep Learning (ML & DL) techniques, present numerous possibilities for the processing of highly complex and noisy data sets (*Bishop, 2006*). Combined with their versatility, in many cases ML and DL approaches have outperformed human experts in a multitude of specialized tasks. Here we present a means of utilizing AIAs for the detection of patterns in 3D spatial data, establishing a method to objectively and efficiently detect discrete levels that human experts may have mistakenly overlooked. This study includes both unsupervised and supervised ML, joined through a hybrid Human-AI collaborative approach in order to establish a final model that can be used to describe the palaeontostratigraphic nature of each site.

The main goal of this study is to establish whether or not ML analysis can be used to quantitatively identify discrete fossiliferous levels in paleontological and archaeological sites based on the study of the spatial distribution of fossils. Specifically, this method has been applied and tested in two Batallones Butte sites (Batallones-3 and Batallones-10) in order to determine if the deposits are homogeneous or heterogeneous with discrete fossiliferous levels.

## MATERIALS AND METHODS

The site of Batallones-3 has undergone systematic field excavations between the years 2001 and 2017, whereas excavations at Batallones-10 started in 2007 and are ongoing. Batallones-3 corresponds to the carnivoran-rich lower part of the hour-glass structure previously mentioned. It is not clear whether an upper part ever existed for this fossil site,

but, if it existed, it was destroyed by slope-erosion processes (this site is located in the slope of the Batallones Butte). In turn, Batallones-10 corresponds to the herbivore-rich upper part of the hour-glass structure and, if present, will have a lower part, yet to be excavated. At both sites, standard fossil vertebrate excavation protocols were followed in the extraction of paleontological remains (*Eberth, Rogers & Fiorillo, 2007*). On-site documentation of excavated remains has primarily consisted of taxonomical and anatomical identification as well as in depth documentation of the spatial distribution of remains. Spatial data obtained thus include standard $x$, $y$ and $z$ coordinates according to their position within a grid, as well as the trend and plunge of elongated fossil remains. Additional data, for future taphonomical studies, were also collected concerning the degree of articulation between anatomical elements, the overall preservation of the remains and the restoration techniques used.

For the purpose of this study, the data used for both supervised and unsupervised ML applications consisted solely in the spatial distribution of remains, including their $x$, $y$ and $z$ coordinates for in-depth 3D analysis. Manageable slices were extracted, similar to the approach proposed by *Canals, Vallverdú & Carbonell (2003)*, 2 for each site across the entire length of the excavated area. In the case of Batallones-10, one 2-m-wide slice was taken across the $x$ and one across the $y$ axis, searching for representative areas with the greatest object densities to capture a global vision of the entire site. In the case of Batallones-3, located in a domically-shaped cave with a debris cone in the middle under an inferred opening, two 2-m-wide slices were obtained along the site's $x$ axis, one on each side of the debris cone.

Once slices were selected, the coordinates of each of the remains were imported into the free R software, x64 v.3.5.1 (www.rproject.org, *R Development Core Team, 2018*), for further analysis.

## Unsupervised machine learning

For initial detection of hidden patterns among the levels of Batallones Butte site, an unsupervised density based clustering algorithm was used. Unsupervised ML algorithms are highly efficient AIAs that are exceptional for their use in tasks such as pattern recognition, anomaly detection, noise and dimensionality reduction, feature engineering as well as generative modeling (*Patel, 2019*). While unsupervised learning performs poorly in specifically defined tasks, their greatest advantage can be seen through their flexibility when applied to versatile datasets for general feature extraction. A popular application of unsupervised algorithms lies in clustering tasks, a component of pattern recognition that is, considered useful for processing and searching for hidden trends in highly noisy datasets.

A non-parametric Density-Based Spatial Clustering of Applications with Noise (DBSCAN) algorithm from the "fpc" R package was used for pattern recognition in 3D spatial data. Proposed as a means of overcoming clustering issues where groupings of samples are not straightforward (*Ester et al., 1996*), DBSCAN is highly efficient at detecting patterns in arbitrary and noisily distributed samples (*Sander et al., 1998*; *Schubert et al., 2017*).

DBSCAN performs through establishing areas of significant point densities, thus providing an efficient means of finding non-parametric patterns among noisy data sets while easily detecting outliers in low-density regions. This is especially useful when used for pattern recognition in spatial studies, where traditional clustering models such as partitioning and hierarchical algorithms tend to focus groupings around a centroid or subsamples in a convex manner. This frequently creates conflict in real-life circumstances.

DBSCAN has proven efficient when applied to 2D, 3D and/or higher dimensional feature spaces (*Ester et al., 1996*; *Sander et al., 1998*; *Schubert et al., 2017*). The algorithm works by separating points within the cluster (i.e., *core points*) from those on the border of the cluster (i.e., *border points*). The differentiation between the two is established mathematically depending on the reachability of neighboring points ($q$) from a core point ($p$). To define this, DBSCAN requires two main hyperparameters, the $\varepsilon$ value defining the neighborhood (a.k.a. Eps) of a point and the minimum number of points (MinPts) that are required to form the cluster (i.e., the density of points). From here DBSCAN defines the neighborhood of $p$ within dataset $D$ as $N_{Eps}(p) = \{q \in D \mid \text{dist}(p, q) \leq \varepsilon\}$, adjusting for border points and the exclusion of noise through $p \in N_{Eps}(q)$ and $|N_{Eps}(q)| \geq$ MinPts (*Ester et al., 1996*). The definition of *density-reachable* points for clustering is thus established if the points can be linked in a chain, whereby $p_1, \ldots, p_n = q$, $p_n = p$ ensuring $p_{i+1}$ is directly reachable through density from $p_i$. Through this definition, border points may not be density reachable if they do not fulfill the core point condition $|N_{Eps}(q)| \geq$ MinPts. The final components that define point density for clustering are established through *density-connected* points, whereby a point $o$ is density-reachable from both $p$ and $q$, essentially connecting both border and cluster points.

Once density has been established within the sample, the assigning of these points into cluster $C$ requires the fulfillment of conditions; (1) $\forall$ $p$, $q$: if $p \in C$ and $q$ fulfill the definition of being density reachable, then $q \in C$, while (2) $\forall$ $p$, $q \in C$: $p$ must be density-connected to $q$. These defined conditions are named *Maximality* and *Connectivity*, respectively (*Ester et al., 1996*). In accordance with the aforementioned conditions, the final definition of noise can be logically established through $\{p \in D \mid \forall i: p \notin C_i\}$.

DBSCAN was thus trained on the entire dataset in an unsupervised manner, establishing an average MinPts value between 3 and 5. These values were intuitively chosen depending on slices with tightly packed densities (MinPts $\approx$ 5) and slices with a lower number of remains (MinPts $\approx$ 3). This abides by general rules recommended by numerous authors (*Sander et al., 1998*; *Schubert et al., 2017*), who define an approximate optimum MinPts = 2 $\times$ n° dimensions in the dataset which is then adjusted depending on the complexity of point distributions and the analyst's knowledge of the domain under consideration (*Sander et al., 1998*; *Schubert et al., 2017*). $\varepsilon$ values were established in accordance with both the correspondent MinPts values and the slices under study. For optimization of this hyperparameter, $k$-distance graphs were plotted according to the nearest neighbor (*Ester et al., 1996*), employing the "elbow" technique to find a $\varepsilon$ value according to both the MinPts parameter and the actual dataset (*Thorndike, 1953*; *Patel, 2019*; *Satopa et al., 2017*). The use of this heuristic, as provided by the "dbscan" R package, can then be adjusted to find the smallest $\varepsilon$ value possible that can define a final model for

training. DBSCAN then establishes clusters within the dataset, separating noise into a separate "0" cluster and creating convex hulls through the use of border points $q$.

## Collaborative intelligence learning

Considering the currently unquantifiable nature of numerous geological features that ML algorithms may be unable to detect, a human-in-the-loop collaborative strategy was proposed for this study (*Kamar, 2016*; *Holzinger, 2016*; *Dellermann et al., 2019*). The objectives of including this hybrid intelligence strategy meant that human interaction could complement the strengths of the pattern detection algorithms (*Simard et al., 2017*; *Dellermann et al., 2019*), whereby creating a bridge between both unsupervised and supervised techniques for model creation.

The Expert-in-the-Loop (EitL) approach adopted here consisted in the revision of the clustered dataset by considering possible underlying geological components that could be generating noise undetectable by the model. Components of this nature could easily include disconformities, geological faults, erosive surfaces, uneven sedimentation or even sedimentation of geological elements such as large boulders that could be separating point-densities that would essentially belong to the same layer.

The strategies employed consequently used EitL (in this case geologist-in-the-loop) interaction to manually correct clustered patterns, assessing which of the clustered groups were geologically separated and assigning these clusters to a new labeled layer. This was performed as objectively as possible through complimenting paleontological data routinely collected from the site during excavation and evaluating the nature of each of DBSCAN's groupings. EitL thus benefits through *Machine Teaching* (MT) in as much as the expert knowledge is used for troubleshooting and debugging (*Dellermann et al., 2019*; also known as a *sense-making* approach). Moreover, the amount of human input for models was monitored in a *collective* manner, using numerous *domain experts* to ensure accuracy (*Dellermann et al., 2019*). In cases where clusters seemed dubious or could not be clearly defined by either the EitL participants or the MT algorithm, these were stripped of their labels and included in the noise (cluster "0") group for objective classification by the trained ML algorithms in the supervised phase of the system. Once revised by the EitL participants, each cluster group was assigned a new label that could be used to train supervised algorithms in the following part of the workflow.

## Supervised learning

Once spatial data points had been passed through the DBSCAN algorithm for initial pattern detection and then corrected by geologist-in-the-loop interventions, supervised algorithms were employed to define the final fossiliferous level models and classify those points in cluster group "0".

Considering the amount of data available, no bootstrapping procedures were deemed necessary prior to supervised training. Each algorithm was thus trained using 70:30% [*train:test*] splits using $k$-fold cross-validation ($k = 10$) in order to ensure the model could efficiently adjust its weights. While some authors propose the use of Spatial Cross-Validation (SCV) for studies regarding geographic data (*Miller, 2004*;

*Brenning, 2012*; *Pohjankukka et al., 2017*; *Lovelace, Nowosad & Muenchow, 2019*), these algorithms were not seen to produce any significant change to the quality of results. Additionally, hyperparameter optimization was performed using a random search loop function programed in R; establishing the best hyperparameter values for each model using a combination of random values until finding the optimum settings (*Bergstra & Bengio, 2012*). These loop algorithms ran for 50 iterations and were then extrapolated and used for the final classification models.

Two primary supervised ML algorithms were used for the final fine tuning of the fossiliferous level models.

- **Support Vector Machines (SVM)**. SVMs map out input vectors into a non-linear high dimensional feature space, using hyperplanes to calculate the degree of separation between samples (*Cortes & Vapnik, 1995*). In order to overcome traditional limitations imposed by linearity, a *kernel* function is used to define the feature space (*Bishop, 2006*). The constructed hyperplane can thus be used as a discriminant classifier decision surface which uses maximized margin or decision boundaries to reduce chances of overfitting (*Cortes & Vapnik, 1995*; *Bishop, 2006*). The consequent hyperplane can then be used to plot the maximized margin separations between samples and can provide a visual means of dividing strata, in an efficient and objectively computed way. For SVM applications the "e1071" R package was used.
- **Random Forest (RF)**. RF can be described as a robust and highly complex form of decision tree which has proven useful in the past for the processing of spatio-temporal data (*Hengl et al., 2018*). The RF algorithms use small random numbers of data set variables rather than the whole dataset, constructing an independent decision tree with each subsampling (*Breiman et al., 1984*; *Breiman, 2001*). The random variable selection is additionally performed using bootstrap aggregation, using a technique frequently referred to as out-of-bag observations. RF is then able to calculate the number of iterations needed to minimize the out-of-bag error. Once the number of trees has been selected, the algorithm averages results to produce a robust classification model (*Breiman, 2001*). For RF applications the "caret" R package was used.

Model evaluations were performed following standardized ML protocol, evaluating the Kappa (κ), Sensitivity, Specificity and Balanced Accuracy values obtained through confusion matrices (*Kuhn & Johnson, 2013*; *Lantz, 2013*). Sensitivity, Specificity and Balanced Accuracy are values derived from evaluations of Type I and Type II statistical errors in proportion with the rest of the calculated confusion matrix. These values are presented as numbers between 0 (poor) and 1 (high performing classifiers). κ values are a further statistical adjustment of the accuracy metric that measure model agreement relative to what would be expected by chance (*Kuhn, 2008*). These values are similarly presented as numbers between 0 and 1 although negative values can (although rarely) occur. Further evaluation of this statistic uses 0.8 as the threshold between poor and powerful classification models. Finally, loss metrics were recorded to evaluate the

predictive power of each model. For this, the Mean Squared Error (MSE) metric was employed:

$$\frac{1}{n}\sum_{i=1}^{n}(t_i - p_i)^2$$

taking into consideration the target ($t_i$) and predicted ($p_i$) values for each classification. MSE values are further interpreted considering AIAs of this type are trained to reduce the error produced in CV sets, therefore the smaller the MSE value the more powerful the predictive model. Model evaluation was performed using the "caret" R package.

### Fine tuning of fossiliferous levels

Once the optimal models were defined, these were used to classify any of the points separated as noise by DBSCAN as well as the clusters that the EitL participants could not objectively define as a part of either fossiliferous level (so as to avoid subjectivity). In order to empirically class any point as a member of a fossiliferous level, the trained SVM and RF algorithms were used to predict the class label of each point. A threshold of 80% security was established as acceptable. Any points that could not be classed by the model with over an 80% predictive decision boundary were thus rejected and therefore not classified into any level in the final fossiliferous levels models.

## RESULTS

### Unsupervised machine learning clustering

DBSCAN clustering proved highly efficient at quickly processing each slice, grouping fossil remains into abundant clusters in both Batallones-3 (Fig. 2) and Batallones-10 (Fig. 3). DBSCAN results in most cases present very noisy profiles, with the detection of large numbers of tightly packed clusters. This is a result of the irregular density patterns detected in the data and probably results from numerous geological and paleontological disconformities that the algorithm is unable to quantify in some areas. Nevertheless, profiles become less noisy towards the extremities of each slice, helping the differentiation between levels, identifying at least three different fossiliferous levels.

On average, DBSCAN identified 166 points (5.6%) as noise in Batallones-10 and 62 (7.76%) in Batallones-3, with the highest levels of noise appearing in Batallones-10 $x$ axis slice ($n = 141$) and Batallones-3 right slice ($n = 56$). For Batallones-3 right slice, this can be attributed to certain areas of the profile having little separation between the layers (Fig. 2C). Nevertheless, excluding noise, most clustered groups seem to follow a trend or pattern that highlights a separation between different fossiliferous levels (see "Collaborative Intelligence Learning").

### Collaborative intelligence learning

While DBSCAN results produced a large number of clusters, detailed analysis through the EitL process identified numerous groupings that belonged to the same fossiliferous level. After careful evaluation, each of these separations could be recognized as products of minute or large disconformities. These can be attributed to the cavity's asymmetrical

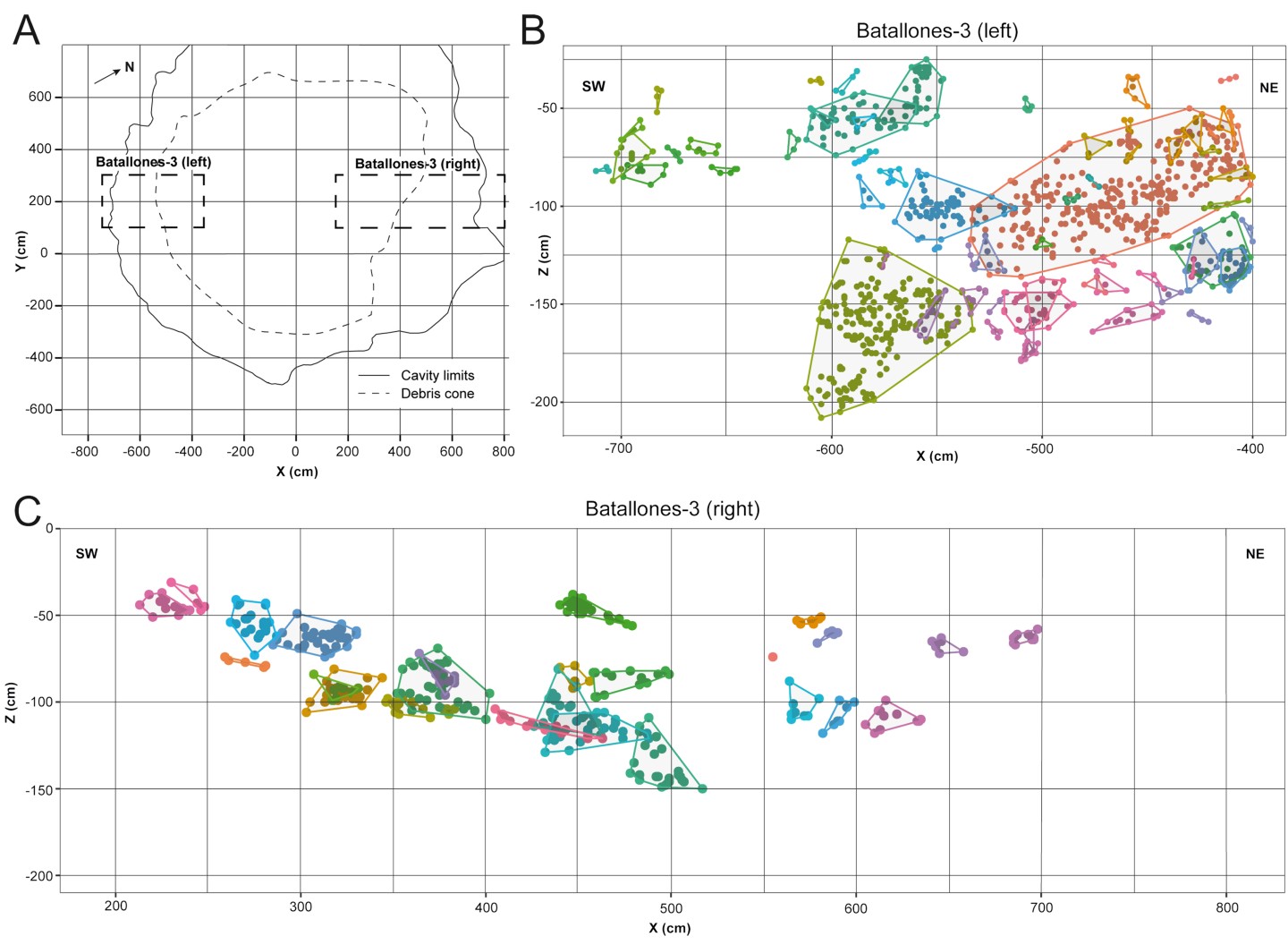

**Figure 2 DBSCAN clustering for the studied slices at Batallones-3, each point representing a fossil remain. Z, depth.** (A) Batallones-3 grid and slice orientation. (B) Batallones-3 left slice clusters (MinPts = 3, ε = 11.5). (C) Batallones-3 right slice clusters (MinPts = 5, ε = 15).

geomorphology at Batallones-10, visible in the *x* axis slice (Fig. 4A) or to a fallen large carbonate block in Batallones-10, as seen in the *y* axis slice (Fig. 4B). This highlights the importance of the geologist-in-the-loop hybrid intelligence system for clarification and unification of clusters.

In other cases, numerous examples can be seen where depositional processes have placed fossiliferous levels closer together such as the sedimentary onlap between levels II and III in Batallones-10 *x* axis slice (Fig. 4A) and Batallones-3 right slice (Fig. 5B). Nevertheless, considering the algorithms' effectiveness processing 3D data, DBSCAN is still able to detect density patterns across the slice's *x*, *y* and *z* axes, thus separating the different levels.

## Supervised learning

Once processed and cleaned by both DBSCAN and the EitL specialists, databases were channeled into supervised algorithms for training. In all cases, each of the databases

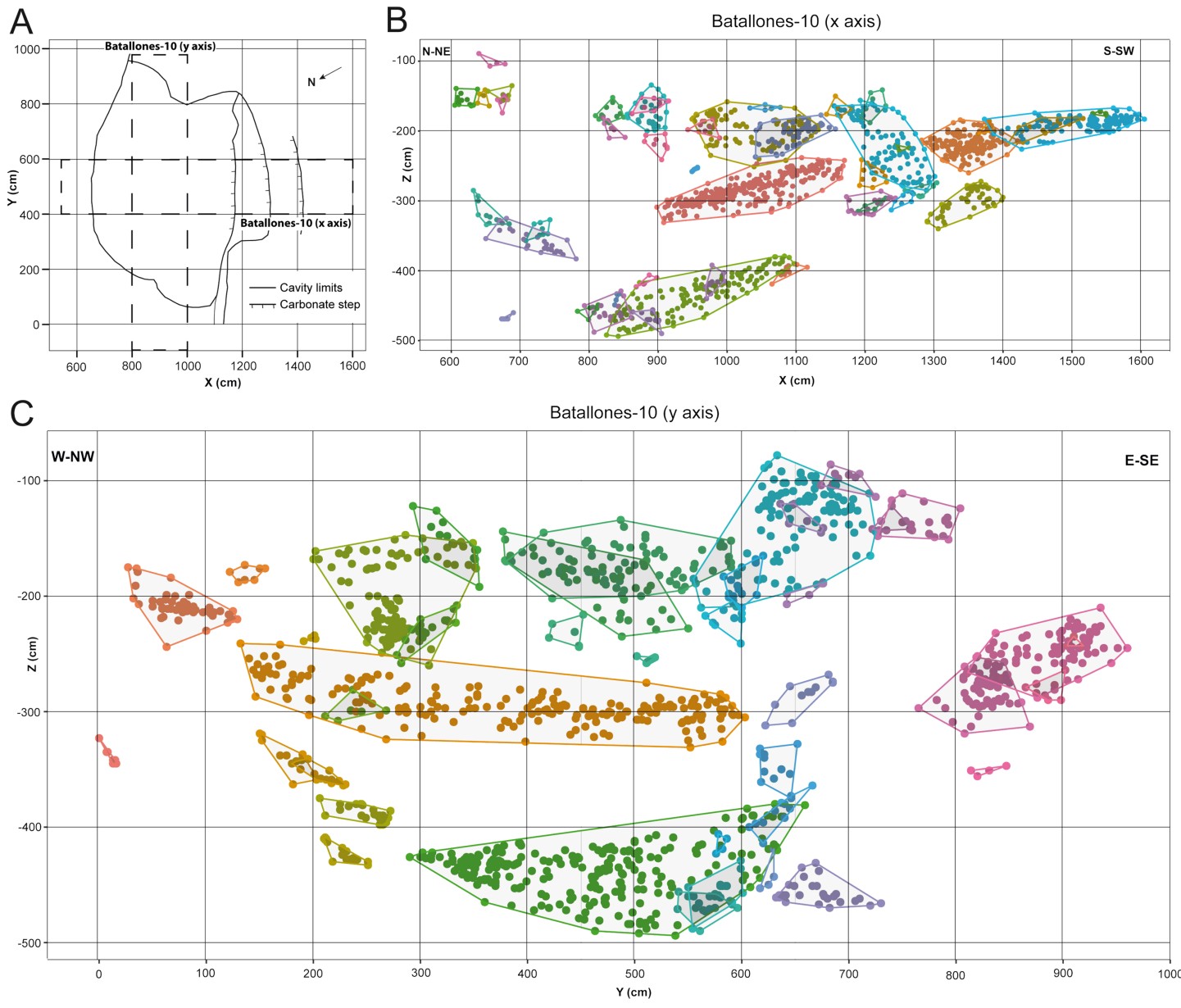

**Figure 3 DBSCAN clustering for the studied slices at Batallones-10, each point representing a fossil remain. Z, depth.** (A) Batallones-10 grid and slice orientation. (B) Batallones-10 *x* axis slice clusters (MinPts = 5, ε = 27.5). (C) Batallones-10 *y* axis slice clusters (MinPts = 5, ε = 25).

generated by unsupervised learning algorithms proved to be highly efficient for the training of their supervised counterparts. Both SVM and RF obtained exceptional results with over 90% accuracy when differentiating between groups and small MSE values on all accounts (Table 1). While SVM proved to have greater balanced accuracy on training sets, RF in general obtained optimal MSE values in testing, especially in the case of Batallones-10. Interestingly Batallones-3 (both left and right slices) seems to create the most amount of confusion, nevertheless on all accounts κ values remain above the acceptable threshold.

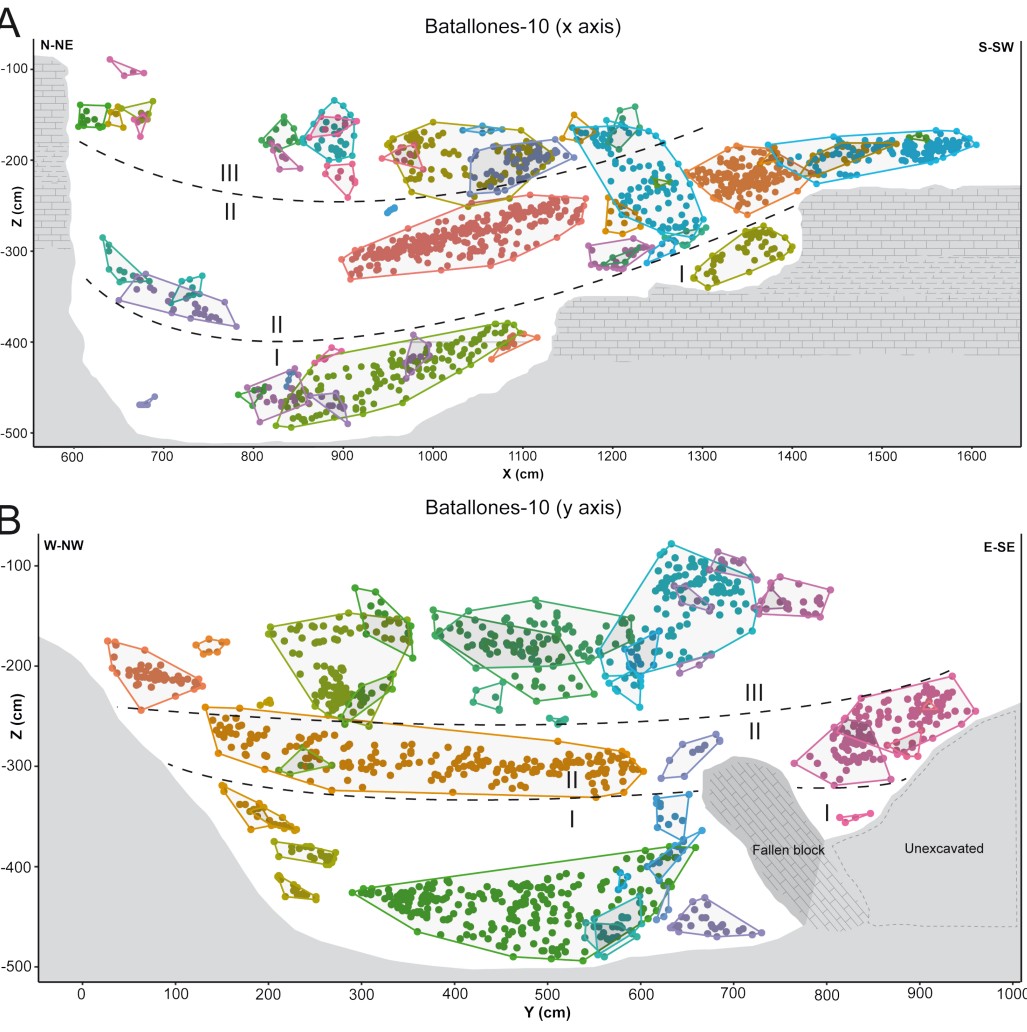

**Figure 4 DBSCAN clustering and geological interpretation. In gray, site limits.** (A) Batallones-10 *x* axis slice clusters and interpretation. (B) Batallones-10 *y* axis slice clusters and interpretation.

Using AIAs to clean the final profiles and de-noise the datasets, both SVM and RF proved to be confident classifiers when making predictions. In most cases, over half the points denoted as noise by the previous phases were successfully classified and assigned into a palaeontostratigraphic level (Table 2). For Batallones-10 *y* axis slice and Batallones-3 left slice, RF outperformed SVM with greater MSE values and success in classifying indeterminable points. Nevertheless, in either case, RF tended to have greater decision making capabilities on some individual points than SVM, even though SVM obtained greater overall MSE in others.

Upon further evaluation of model performance, SVM maximized margin decision boundaries (dotted line in Fig. 6) are seen to expand or contract closer to the identified sedimentary onlap areas (in both Batallones-3 and Batallones-10) while an increase in MSE is observed for these particular areas as well (Table 2).

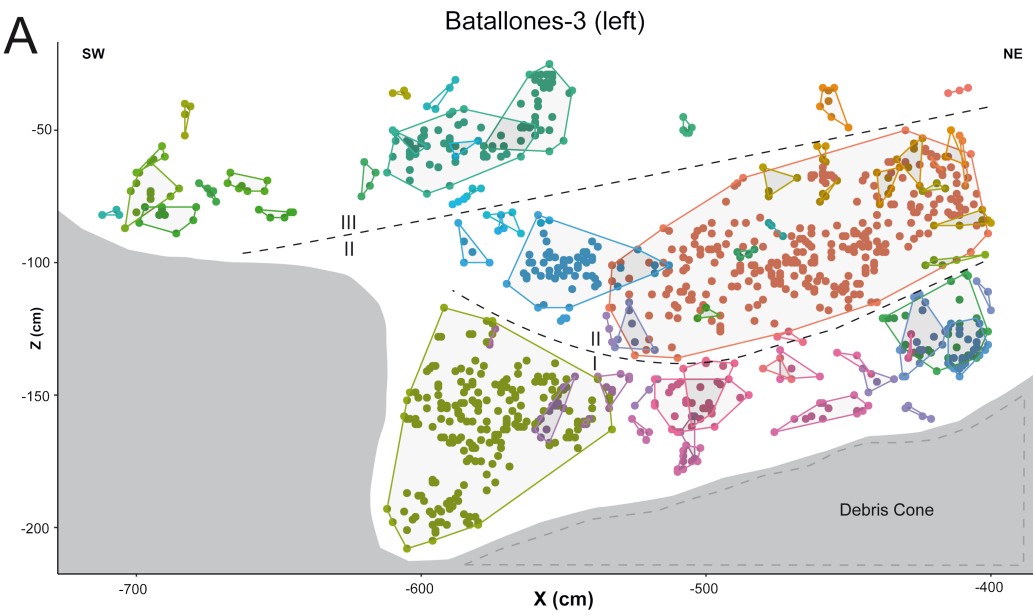

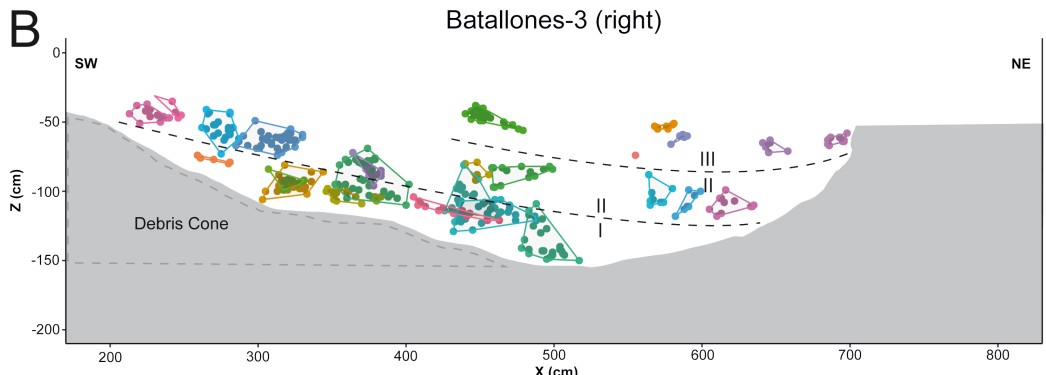

**Figure 5 DBSCAN clustering and geological interpretation. In gray, site limits.** (A) Batallones-3 left slice clusters and interpretation. (B) Batallones-3 right slice clusters and interpretation.

## Fine-tuned models

Once the data had been processed by each of the phases of the workflow, a final fine-tuned model of the fossiliferous levels was created.

The fine-tuned model produced for Batallones-3 shows three discrete fossiliferous levels on either side (left and right) of the debris cone (Fig. 7). The left profile (Fig. 7A) is laterally less extensive (~3 m) than the right profile (Fig. 7B), which extends over 5 m. All identified levels in both profiles dip from the debris cone outwards, with dip angles decreasing towards the outermost parts of the cavity and towards the top of the infilling, with Level 3 dipping only slightly in Batallones-3 left profile (Fig. 7A) and nearly horizontal in the outer part of the Batallones-3 right profile (Fig. 7B).

Batallones-10 fine-tuned model shows three different discrete fossiliferous levels (Fig. 8). A first level (Level I), dips northward (Fig. 8A) and towards the East and West dips

**Table 1 Evaluation metrics of each ML algorithm on test sets, as derived from confusion matrices in File S1.** Confidence Intervals (CI) are established for the accuracy metric.

|  | Random forest | | | | Support vector machine | | | |
|---|---|---|---|---|---|---|---|---|
|  | Batallones-10 | | Batallones-3 | | Batallones-10 | | Batallones-3 | |
|  | $x$ axis | $y$ axis | Left | Right | $x$ axis | $y$ axis | Left | Right |
| Kappa | 0.99 | 0.99 | 0.97 | 0.97 | 1.00 | 1.00 | 0.99 | 1.00 |
| Lower CI | 0.98 | 0.98 | 0.93 | 0.93 | 0.99 | 0.99 | 0.96 | 0.96 |
| Accuracy | 0.99 | 0.99 | 0.98 | 0.98 | 1.00 | 1.00 | 1.00 | 1.00 |
| Upper CI | 1.00 | 1.00 | 1.00 | 1.00 | 1.00 | 1.00 | 1.00 | 1.00 |
| Sensitivity | 0.99 | 1.00 | 0.98 | 0.98 | 1.00 | 1.00 | 1.00 | 1.00 |
| Specificity | 1.00 | 1.00 | 0.99 | 0.99 | 1.00 | 1.00 | 1.00 | 1.00 |
| MSE | 7.73E−06 | 3.39E−05 | 1.15E−03 | 1.15E−03 | 7.76E−05 | 6.84E−04 | 6.07E−05 | 9.37E−04 |

Note:
 MSE, Mean squared error.

**Table 2 Results obtained from ML algorithm for the classification of points considered indeterminable after both MT and EitL intervention.**

|  |  | Batallones-10 $x$ axis | Batallones-10 $y$ axis | Batallones-3 left | Batallones-3 right |
|---|---|---|---|---|---|
| SVM | MSE | 3.45E−03 | 1.21E−02 | NA | 7.11E−03 |
|  | Determinable | 113 | 15 | 0 | 45 |
|  | Indeterminable | 28 | 10 | 6 | 11 |
| RF | MSE | 4.01E−03 | 6.17E−05 | 3.18E−02 | 8.88E−03 |
|  | Determinable | 106 | 23 | 4 | 39 |
|  | Indeterminable | 35 | 2 | 2 | 17 |

Note:
 MSE values represent the confidence of classifications.

towards the central part of the cavity, adapting to the cave limits (Fig. 8B). Level II conformably overlies Level I in the southern section, dipping towards the North and in the northern, eastern and western cavity limits dips towards the central part of the cavity (Figs. 8A and 8B). Finally, Level III is practically horizontal, with gentle folding in the outermost limits of the cavity (Figs. 8A and 8B).

# DISCUSSION

The methodology proposed in this study, based on artificially intelligent systems, has been deemed effective in quantitatively identifying discrete fossiliferous levels in paleontological and archaeological sites based on the study of the spatial distribution of fossils. This new, more quantitative approach could be considered a valuable substitute for previous qualitative techniques that have been applied in the past.

Evaluating variable importance through testing for mean decrease in node impurity for RF algorithms (*Louppe et al., 2013*) explains how a boost in the weight of the third dimension of each slice helps support the differentiation between levels (Fig. 9). Batallones-3, for example, shows a shift in the value of the $x$ axis between the left-hand and

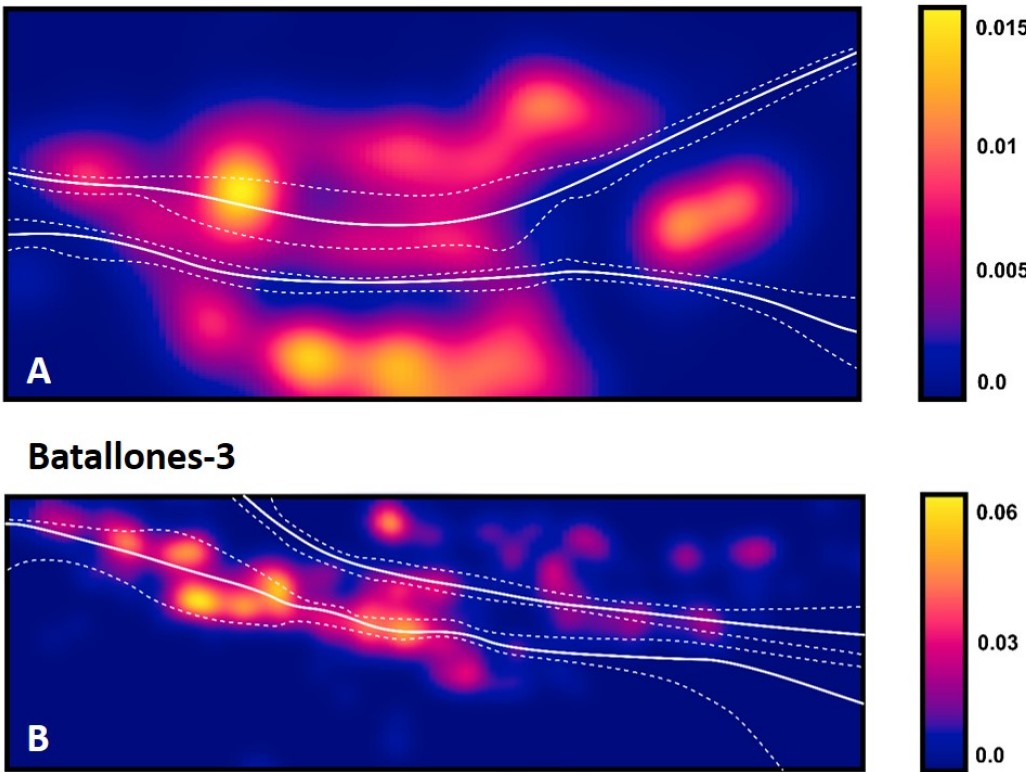

**Figure 6 Examples of SVM maximized margins (dotted) and decision boundaries (solid line), plotted against calculations of material densities.** Density values are reported as the number of points per square unit (cm), per quadrat. (A) Batallones-10 *y* axis slice. (B) Batallones-3 right slice.

right-hand sides of the debris cone, having more weight in the final decision making capabilities of RF than the y dimension. Data of this nature thus supports the need for 3D over 2D analysis in order to ensure an efficient detection of patterns that 2D data may not be able to reveal.

Fine-tuned models produced for Batallones-3, in the lower part of a hypothetical hourglass-shaped cavity, clearly show three discrete levels on either side of the debris cone (Fig. 7). Cave asymmetry, as explained in *Calvo et al. (2013)*, is clearly visible in these profiles, with greater cave development towards the Northeast, in Batallones-3 right profile (Fig. 7B). Batallones-10 fine-tuned models, in the upper part of the hourglass-shaped cavities, show another three discrete fossiliferous levels (Fig. 8). Similar geological structures were observed by *Calvo et al. (2013)* in Batallones-9, located 50 m to the north of Batallones-10.

These newly discovered levels in both the lower levels (Batallones-3) and upper levels (Batallones-10) of the hourglass-shaped cavities of Batallones Butte site support the preliminary inferences made by *Martín Escorza & Morales (2005)* about the possibility of discrete fossiliferous levels within Batallones-1. Careful revision should therefore be carried out considering previous paleoecological, palaeoenvironmental and taphonomical

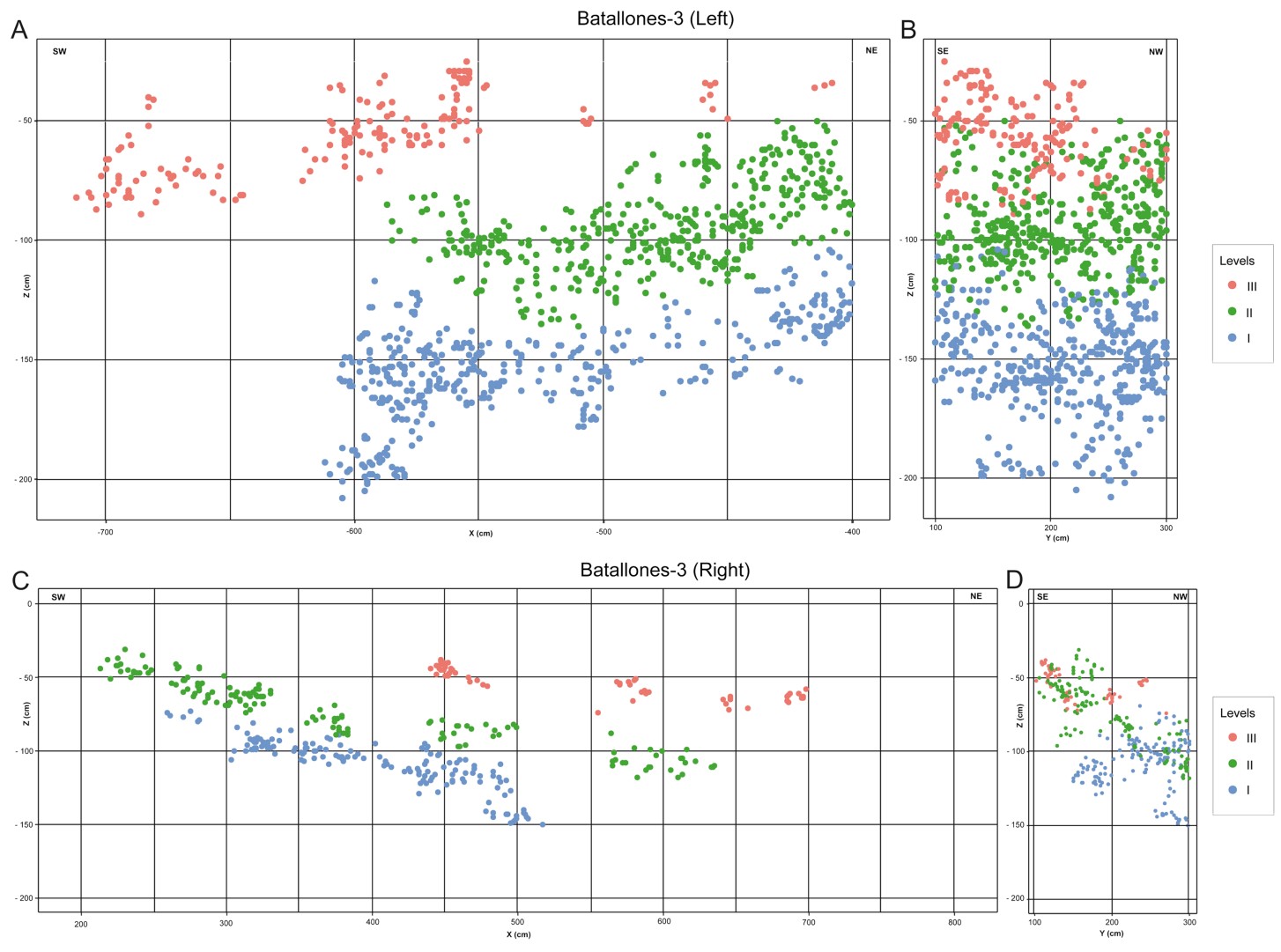

**Figure 7 Fine-tuned fossiliferous level models.** (A) Batallones-3 left slice. (B) Batallones-3 left slice associated 2-m-wide perpendicular slice. (C) Batallones-3 right slice. (D) Batallones-3 right slice associated 2-m-wide perpendicular slice.

analyses. As of this point, future studies at Batallones-3 and Batallones-10 should consider each of these individual discrete levels, instead of for the whole upper or lower part of the cavity.

As previously explained, Batallones-3 deposits and fossils are located in the lower part of the cavity and the fossil site is considered exhausted as the encasing rock was already found, for this reason, the three identified levels can be considered definitive. In the case of Batallones-10, the encasing rock has not yet been found and therefore fossil material recovered in future excavations may reveal additional discrete and separable fossiliferous layers. Thus, the increase in the number of levels at Batallones-10 cannot be discarded.

Future studies should analyze the embedded faunal assemblages separately for each of the described levels rather than the entire upper or lower part of the cavity, paying special attention to micromammal remains in order to establish chronological intra-site

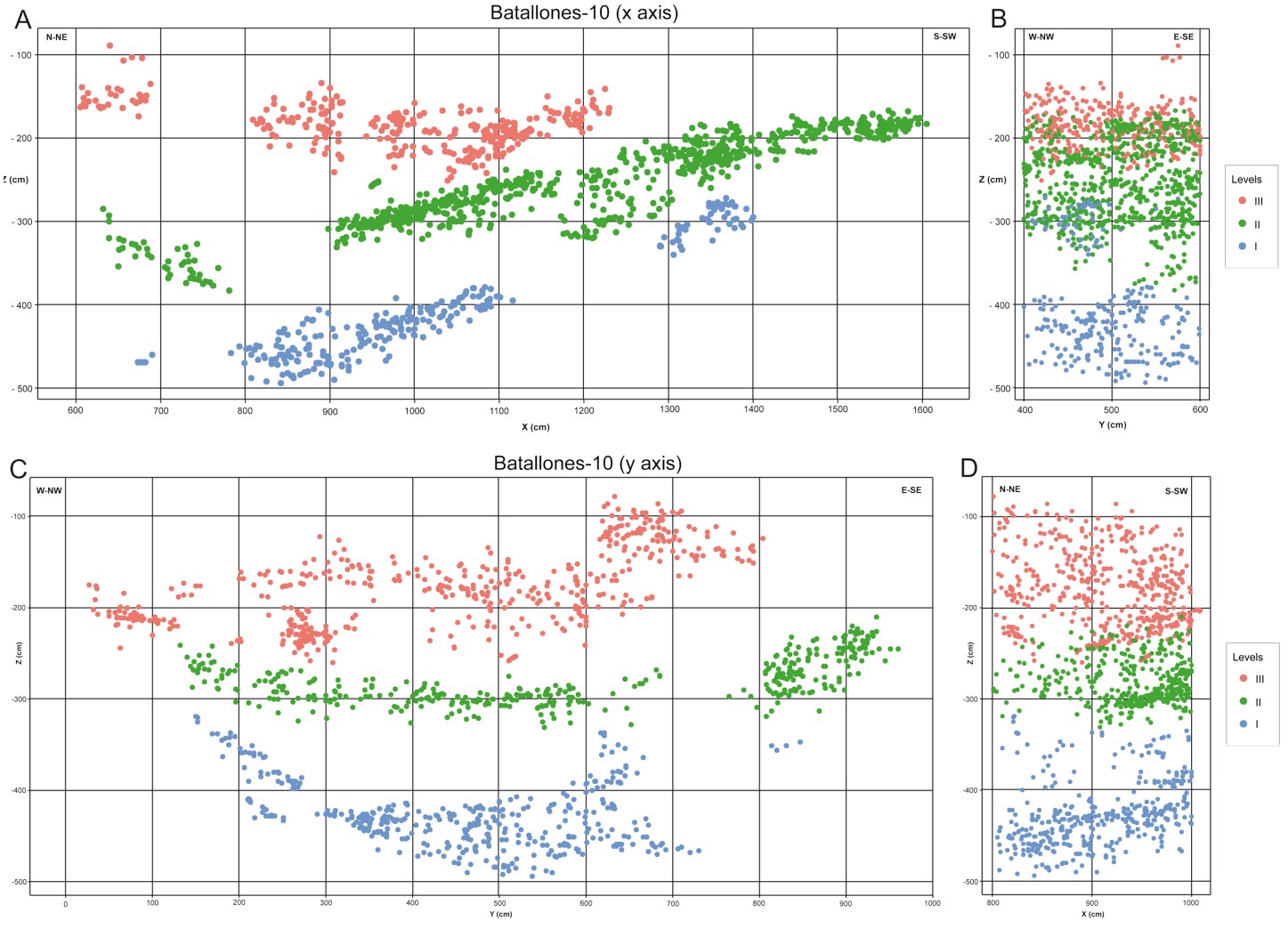

**Figure 8  Fine-tuned fossiliferous level models.** (A) Batallones-10 *x* axis slice. (B) Batallones-10 *x* axis slice associated 2-m-wide perpendicular slice. (C) Batallones-10 *y* axis slice. (D) Batallones-10 *y* axis slice associated 2-m-wide perpendicular slice.

relationships, such as those inter-site relationships described by *López-Antoñanzas et al. (2010)*. These detailed micromammal studies would greatly refine the geochronological framework of the Batallones Butte sites.

The overall workflow of this hybrid intelligence system has proven effective in detecting the presence of 3 levels in both Batallones-10 and Batallones-3. While in some slices the detection of these levels is clearer than in others, the use of two representative, thick and potent slices for each site helps empirically support their identification. Through the implementation of unsupervised algorithms for the initial detection of patterns, expert (geologist)-in-the-loop interactions for sense making and the final tuning of profiles using supervised algorithms, the overall system can be used to find areas where identification of fossiliferous levels is clearer. This can then be extrapolated for the supervised classification of the rest of the site.

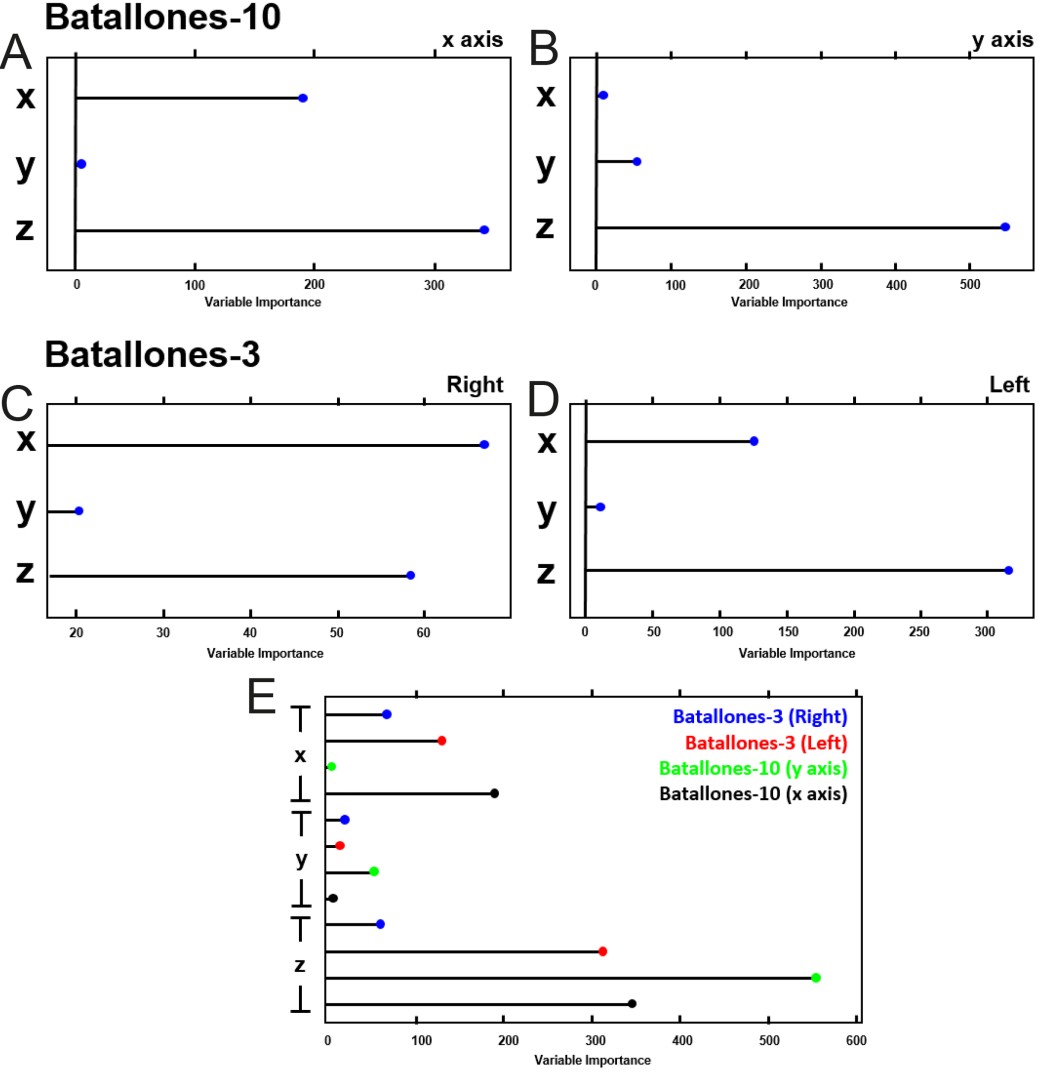

**Figure 9 Variable importance plots for the RF model.** Each panel represents the weight each variable has on the decision making capabilities of the RF model, indicating the importance in some cases of the use of 3D data for stratigraphic model definition. (A) Batallones-10 *x* slice. (B) Batallones-10 *y* slice. (C) Batallones-3 right slice. (D) Batallones-3 left slice. (E) Comparison of all slices.

Moreover, while some isolated points suspiciously appear to be classified into one layer and can be arguably considered out of place, future efforts should try to combine more data for the fine tuning of teaching and training processes. These could include taxonomic or taphonomic variables (*Brain, 1981*; *Domínguez-Rodrigo et al., 2018*), data regarding object density (*Kreutzer, 1992*; *Lam et al., 1998*, *2003*; *Lam, Chen & Pearson, 1999*), weight and size (*Bunn et al., 1980*; *Brain, 1981*; *Bunn, 1987*; *Bunn & Pickering, 2010*; *Domínguez-Rodrigo et al., 2018*), as well as orientation and plunge patterns (*Woodcock, 1977*; *Woodcock & Naylor, 1983*; *Lenoble & Bertran, 2004*; *Domínguez-Rodrigo & García-Pérez, 2013*).

While models and systems without humans-in-the-loop would be an optimal solution for the future (*Holzinger, 2016*), numerous geological factors that need extensive investigation for modeling are required before this can be achieved. Likewise, a number of underlying depositional and post-depositional processes may be generating confusion that machines are unable to understand or process for the time being. Such is the case for the confusion generated amongst algorithms caused by sedimentary onlaps, causing fossiliferous levels to lie closer to each other, at both Batallones-3 and Batallones-10. This may, however, be solved by more complex AIAs and the use of Deep MT systems, such as clustering AIAs using auto-encoders (*Xie, Girshick & Farhadi, 2016*; *Guo et al., 2017*; *Mrabah et al., 2019*; *Yang et al., 2019*), or those used for reinforcement learning tasks (*Lake et al., 2014*; *Mnih et al., 2015*; *Holzinger, 2016*; *Simard et al., 2017*). Efforts should therefore be made to investigate the effects of these numerous geological components on pattern detection algorithms.

## CONCLUSIONS

This study presents a novel use of Artificially Intelligent Systems for the quantitative identification of discrete fossiliferous levels in paleontological and archaeological sites based solely on the study of the spatial three-dimensional distribution of fossil remains. These results have been able to reveal new discoveries in the Batallones Butte site inner structure, including three discrete levels at both Batallones-3 and Batallones-10.

The two lowermost levels of Batallones-3, Level I and Level II, dip outwards from the debris cone towards the outer limits of the cave, becoming progressively more horizontal, whereas the uppermost level, Level III is sub-horizontal. On the other hand, Batallones-10's lowermost Level I dips northward into the cavity and adapts laterally to the cave limits whereas Levels II and III also show northern dips but are more influenced by cave limits and dip towards the center of the cavity in the outermost areas.

Through these discoveries, AIAs have been proved to be a highly efficient and objective means of detecting spatial patterns in paleontological sites. Unsupervised and unsupervised techniques can greatly reduce (but not eliminate) the amount of subjectivity in discrete fossiliferous level identification. The possibilities provided by the combination of these algorithms and expert-in-the-loop systems are multiple, however it is important to point out that in order for these methods to be widely used, a need for further experimentation and investigation is essential. This should include an increase in the data used to feed these AIAs. For example, the incorporation of taphonomic features of the remains in the analyzed fossil may provide an even finer tuning of these models.

## ACKNOWLEDGEMENTS

We would like to thank the TIDOP Group from the Department of Cartographic and Land Engineering of the Higher Polytechnics School of Avila, University of Salamanca, especially Diego González-Aguilera and Miguel-Ángel Maté-González for their support.

We are very grateful to Robert Anemone, João Coelho and an anonymous reviewer for their insightful comments which helped greatly improve this article.

### Funding

This research has been funded by Project PGC2018-094122-B-100 of the Spanish Government (Science and Innovation Ministry). David Manuel Martín-Perea was funded by an FPI Predoctoral grant BES-2016-079460 from the Spanish Government associated to Project CGL2015-6833-P. We acknowledge support of the publication fee by the CSIC Open Access Publication Support Initiative through its Unit of Information Resources for Research (URICI). The funders had no role in study design, data collection and analysis, decision to publish, or preparation of the manuscript.

### Grant Disclosures

The following grant information was disclosed by the authors:
Spanish Government (Science and Innovation Ministry): PGC2018-094122-B-100.
FPI Predoctoral: BES-2016-079460.
Spanish Government: CGL2015-6833-P.
Unit of Information Resources for Research (URICI).

### Competing Interests

The authors declare that they have no competing interests.

### Author Contributions

- David M. Martín-Perea conceived and designed the experiments, performed the experiments, analyzed the data, prepared figures and/or tables, authored or reviewed drafts of the paper, and approved the final draft.
- Lloyd A. Courtenay conceived and designed the experiments, performed the experiments, analyzed the data, prepared figures and/or tables, authored or reviewed drafts of the paper, and approved the final draft.
- M. Soledad Domingo analyzed the data, authored or reviewed drafts of the paper, and approved the final draft.
- Jorge Morales analyzed the data, authored or reviewed drafts of the paper, and approved the final draft.

### Data Availability

Confusion matrices are in File S1. Raw coordinate data and the revised R script used are also available as Supplemental Files.

### Supplemental Information

Supplemental information for this article can be found online at http://dx.doi.org/10.7717/peerj.8767#supplemental-information.

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
