# Peer review of "Application of artificially intelligent systems for the identification of discrete fossiliferous levels"

_PeerJ, doi:10.7717/peerj.8767_

## Round 0.1 · original submission · Major Revisions

The three referees have all found merit in the work and suggest relatively modest revision. The comments from referee 3 seem particularly valuable and may deserve special attention. I hope that you are able to use the referees' comments to revise your article.

·

Basic reporting

This paper is well written in clear professional English with just a few exceptions that can be easily fixed. Line 89, I think the authors mean to say “restoration”. On lines 118-119, it might be better stated that “At both sites, standard fossil vertebrate excavation protocols were followed in the extraction of paleontological remains (Eberth et al. 20007)”. On lines 123-124: “Additional data were also collected concerning the degree…”. The final sentence on lines 161-162 should be revised…while “conflictive” is apparently an English word, it is not an often used word, and I think the sentence can be phrased more clearly by using an alternative form, perhaps some form of the verb “to conflict”. Lines 165-166…”The algorithm works by separating points within the cluster from those on the border of the cluster…”. Lines 177-178…”The final components that define point density for clustering ARE established…”. Line 221…” separated and assigning these clusters to a new labelled layer”. Line 224-226…”in as much as the expert knowledge is used for troubleshooting and debugging (Dellerman et al., 2017:70: this is also known as a sense-making approach.” Lines 254-255…”These loop algorithms ran for 50 iterations and were then extrapolated and used for the final classification models.” Line 258 I would drop “These consisted in:”. Line 273…”The RD algorithms USE small random numbers”. Line 282…use either “Model evaluations were performed” or “Model evaluation was performed”. Line 288…”that measure model agreement”. Lines 316-317…”This is a result of the irregular density patterns detected in the data and probably results from numerous geological and paleontological…”. Please make clearer the meaning of the sentence beginning with “Nevertheless” on lines 318-320. Line 327 “highlights”, and please edit/clarify the sentence on Lines 324-325 beginning with “For Batallones-3 right slice”. Line 331…”a large number of clusters”. Line 333…”could be recognized as products of”. Line 342, “effectivity” is not the word you want here (not sure if it is a word)…please rephrase. Line 435…”rather than the entire upper or lower part of the cavity”. Lines 429-432 “In the case of Batallones-10, the encasing rock has not yet been found and, therefore fossil material recovered in future excavations may reveal additional discrete and separable fossiliferous layers”. Thus, the increase in the number of levels at Batallones-10 cannot be discarded.” Line 436…”such as those described by Lopez-…” Line 477-478…”including three discrete levels at both Batallones-3 and Batallones-10.” Line 493…”an even finer tuning of these models”.
The Intro and Background sections are reasonable, although contrary to the claim made on lines 91-92, a number of vertebrate paleontologists have been developing AI approaches to site location (including Paul Barrett, Amy Chew and Kate Oheim, RL Anemone, CW Emerson, GC Conroy) and this work should be referenced in this paper.
Figures are nicely put together and relevant and raw data are supplied along with R code.

Experimental design

In most respects, the manuscript fits within the scope of Peer J, but there are some issues worth considering. While the investigation is rigorous in terms of experimental methods and the methods utilized are highly appropriate, some important aspects seems to be missing. The nature of the fossiliferous levels that the paper explores is not clearly described. Are the authors attempting to identify biostratigraphic layers that differ in time from within a site, or are they trying to identify assemblages that have different depositional or post-depositional histories? What meaning do the authors ascribe to the purported “discrete fossiliferous levels” is never made clear. As I understand it, the two sites under consideration were never separated into discrete fossil-bearing levels. Therefore, I’m unclear what the training for the supervised classification was based on and I’m puzzled about the absence of a confusion matrix to identify how well the algorithm worked. I imagine that the authors used the DBSCAN results and geological interpretations presented in Figures 3 and 4 as the basis for identifying 3 discrete levels, and then trained the supervised classifiers based on these results. Is this correct? If so, I’m unsure about the process of drawing the dotted lines in these plots to determine the existence of 3 levels…this seems arbitrary (why not 2 or 4 levels?), but perhaps the authors can better explain this to my satisfaction. The need for a confusion matrix remains in order to evaluate the operation of the predictive algorithms.

Validity of the findings

I believe that many of my comments in the previous section are perhaps relevant to this section too. The authors need to do a better job of convincing me that there is nothing arbitrary about their conclusion of 3 and only 3 discrete layers at Batallones -3 and 10, as well as their understanding of what these layers mean for interpretation of the sites under consideration.

Additional comments

I commend the authors for their innovative application of sophisticated AI approaches to the problems of taphonomy and biostratigraphy and I’m convinced that this work establishes a proof-of-principle for this kind of work. I wonder if a better test of their methods might be to try to reproduce a clearly determined, a priori biostratigraphic zonation (based on standard approaches) at some site in order to see if their AI approach could replicate it. Or alternatively, they could compare the results they present in this manuscript to a standard zonation of the same site in a blinded fashion. The reader would then be able to more easily understand how well the new approach works in relation to the previous methods used.

Reviewer 2 ·

Basic reporting

This article is generally well written with only a few minor grammar or spelling issues. Most style guides recommend that terms like "Machine Learning" and "Artificial Intelligence" should only be capitalized when they are a proper noun, such as when they are in a journal or conference title. The article is structured according to the Peerj guidelines. The references are generally ok (some are separated by a blank line, others are not), but more descriptive background on the characteristics of the fossils is needed. The distinction between the upper herbivores and lower carnivores in the two lobes of the caves is understood, but do the three modeled layers at each site contain different species? Much more background on the taphonomy and paleoenvironment at the two sites would give the reader a better idea of the problem domain. A simple map of the two localities that shows how the slices are oriented with respect to the local Cartesian coordinate system is needed to fully comprehend the spatial configuration of the fossil point cloud. Other issues with figures are that the font size in Figure 2 is very small—this should probably be split into two separate figures at larger scales (one for each locality) as was done in Figures 3, 4, 6, and 7. Why is only the right slice of Batallones-3 depicted in Figure 4? In Figure 5, the legends need units. All of the Importance axes of the Figure 8 plots should be on the same scale to facilitate comparison. The DBSCAN unsupervised clustering algorithm requires a minimum points parameter along with epsilon, a maximum distance value. The latter is determined using the “elbow” of k-distance graphs. The epsilon parameter used in the final model needs to be listed, and the appropriate k-distance graph from which this value was derived should be included as well. Line 375 incorrectly refers to Figure 6B as the left profile. Lines 404-410 in the Discussion section repeat what was said in lines 371-379. Similarly, lines 412-417 repeat lines 381-386.

Experimental design

The unsupervised clustering of the 3D locations of fossils in the two sites is fairly well explained (with the exception of the epsilon parameter noted above) but the collaborative learning discussion needs much more detail. In my reading of the text, a number of geologists collaborate to draw the layer boundaries between clusters along the slice (x-direction for Batallones-3) and across the 2 meter slices. In Batallones-10 the layer boundaries are drawn between clusters that extend along the longitudinal axes of the x and y slices. It is never made clear why the slices are parallel at Batallones-3 and orthogonal at Batellones-10. 70 percent of the points were then used to train a support vector machine or random forest supervised classification algorithms. These results had very high accuracy statistics, but one has to wonder what the results would be if the classifiers were trained on 30 percent of the points and assessed on the larger portion. It isn’t really made clear exactly how the supervised classifications moved the layer boundaries or otherwise improved on the expert-in-the-loop. The formula for mean squared error on line 294 is incorrect (missing the squared exponent). The fine tuning of the levels is not explained sufficiently. Was this another supervised classification of the noise points or was it manual reassignment of the points like the expert-in-the-loop approach?

Validity of the findings

It was helpful having the R script to get a better idea of the methodology.
Lines 104-106 state that the goal is to establish whether or not machine learning can quantitatively identify fossiliferous levels in paleontological and archaeological sites based on the spatial distribution of fossils. Lines 487-488 in the conclusions state that artificial intelligence algorithms were shown to be an objective means of automating spatial pattern detection. The DBSCAN clusters certainly provide a valuable quantitative input to the expert geologists who identified the layers, but the entire process is not really automated. The authors admit this on lines 458-469 and the conclusion that taphonomic features can fine tune the model is a point well taken. It may be a good thing to adjust the goal and conclusions to emphasize how unsupervised and supervised techniques can reduce (but not eliminate) the amount of subjectivity in decisions made using the expert-in-the-loop process.

·

Basic reporting

This paper introduces a new quantitative method for classifying fossiliferous levels, using the following pipeline: 1 – clustering of xyz coordinates (dbscan), 2 – domain-expert intervention (hard-coded grouping of clusters), 3 – classification (svm, rf). This approach might be useful when there are no clear stratigraphic horizons. The paper is well structured, figures are clear, data is available, and the code provided is documented. Overall, the manuscript is clearly written but I have worries over some use of terminology, and I believe there are many things that can be improved in that regard.
The introduction is strong, providing important background and contextualization for the late Miocene fossil sites. There is however a large gap in the literature references of AI applications to palaeontology and a quite strong statement (lines 91-93) that needs revision. Palaeontologists have been using machine learning for a myriad of problems, there is not just one exception! Just recently, Püschel et al (DOI: 10.1098/rsif.2018.0520) used multiple supervised classification (and other algorithms) to tackle a specific hypothesis (concerning functional/morphological data) in palaeontology – I am sure there are many more papers like this. But far more relevant to the authors are the several papers on paleontological remote site detection using neural nets and other approaches. See Wills et al. (DOI: 10.1016/j.palaeo.2017.10.009) a simple MDS + LDA pipeline (but still); Block et al (DOI: 10.1371/journal.pone.0151090) uses an “ensemble”-like tactic combining Bioclim, MaxEnt, GLMs and logistic regressions; and see also the several papers of Conroy, Anemone, Emerson, and others (ca. 2011-2015) using unsupervised and supervised methods to find fossil sites in Utah and Wyoming. The recently published book “New Geospatial Approaches to the Anthropological Sciences” also has some chapters about these machine learning approaches for fossil site detection from satellite images. I consider these additions are essential to show how your work fits into the broader field of AI applications to palaeontology.

Experimental design

The documentation of the spatial distribution of remains is the chosen method of data acquisition. Thus, the dataset is exclusively composed of non-labelled triplets of 3D-coordinates. The authors mention the collection of the following variables: taxon/id; bone element; degree of articulation; preservation; and so on. Nevertheless, such variables were not used in any step of the statistical modelling pipeline. It is not very clear why.
First step proposed by the authors is to apply the popular outlier-detection algorithm, dbscan. In line 152 the authors mentioned they used the dbscan package implementation, however in their code the fpc package implementation (which is quite different) is used instead. This needs clarification, since it looks like the dbscan package was only used to find hyperparameters through the visual/manual elbow method. It can also confuse amateur R users, e.g. method = "hybrid" (in the code provided) works in the fpc::dbscan but in the dbscan package implementation it is an unknown parameter leading to an output error. In line 201 you mentioned dbscan (step 1) “removing noises” – however in the code you provided this is manually hard-coded in the end of step 2. Maybe rephrase it to “identifying noises” which is what dbscan does by assigning noisy rows to cluster 0.
Step two is a manual labelling step, effectively clustering all the clusters into less clusters. In my view, it is described in a very glorified and puzzling way. But before going into that, there is a problem of designations. I believe it is less confusing if authors would instead stick to just one designation (or explain why there might be many and then stick with one). In my view, this seems ultimately just a simple step for reorganizing labels, as dbscan alone did not match the desired results. Yet, we are confronted with the following terminology: “Collaborative Intelligence Learning”; “human-in-the-loop collaborative strategy”; “hybrid intelligence”; “Expert-in-the-Loop (EitL)”; “geologist-in-the-loop”; “Machine Teaching (MT)”; “sense-making approach”; “MT-DBSCAN”. I suggest authors to keep it simple and thus clearer. Lines 232-234 say that with this hybrid approach noise can finally be identified, but previously in line 201 noises were removed. Very contradictory, please clarify. If the problem step 2 is trying to solve is dbscan’s final output over-generating clusters, I think hdbscan developed by Campello, Moulavi, and Sander (2013) might work as an alternative to step 1+2. If the authors have not tried this before, I suggest it might be worth a try, because relabelling by geologists as a mandatory mid-step in the pipeline makes it harder to streamline the method and generalize it to other sites.
Final step is a supervised classification of the labels reorganized by the experts in the previous step. Train/test split is correctly done, as well as 10-fold cross-validation. However, authors describe at length they used a special version of the k-fold CV algorithm for spatial data (lines 243-250). Notwithstanding, the code instead provides a standard implementation of the CV algorithm from the caret package: trainControl(method = "cv", repeats = 10). Also, just for clarity, repeats= control repeats, folds are controlled by the “number = 10” argument. Moreover, the following lines (251-255) described a systematic hyperparameter search that I also did not find in the code provided. Lines 268-269 mentioned e1701 and kernlab for applying svm, but in the code caret was used instead. Line 282 mentions deep learning again, and I really fail to see how DL is relevant for this paper’s methods. Also, DL is not a different thing from ML, it is a subset of it. Lines 302-307 mentioned an 80% security threshold to allow correct identification of the unknown datapoints during the predict(), which is also not available in the code. Thus, the code to generate Table 2 is also not available.

Validity of the findings

My main concern in terms of the validity is that many parts of the methods were described in ways that do not fully correspond with the code provided. It seems like right now the experimental design and results are not fully reproducible even though all the data was provided. Which I find extremely troublesome. The results section also has some details I would like to see clarified. It is never explained how the proper “corresponding palaeontostratigraphic level” is defined for noisy points and how the success can be measured if the true label is not known a priori (lines 358-360). If this is achieved based on a strict prediction that excludes the possibility of the “noise”-label, it should be stated in a straightforward manner as this means you do not have a statistical test of success, just a prediction. I found the use of the term “cleaned” (line 372) to not be the best choice. After all, supervised algorithms are classifying/discriminating the data, not “cleaning” it (data cleaning as a technical term is usually for pre-processing steps, before modelling, like removing NAs, etc.).
Discussion/conclusions do a good job at highlighting the relevance of this work and providing a framework for future directions. If the new stratigraphic horizons make sense, future work can refine geochronological relationships using e.g. biostratigraphy of micromammals. However, before that, the pipeline needs some improvements. Maybe the next steps are indeed to add more variables as suggested by the authors, specifically containing taphonomic, geological and faunal information to refine this approach so it can be really automated or close to automation. Trying different unsupervised approaches might also improve the results in the first step of the pipeline. Ideally so we do not even need step 2.
I am glad that this approach was useful to better understand the boundaries of stratigraphic horizons at the Batallones sites 3 and 10. This is undoubtedly the strong point of the paper, and it shows the method described has clear promise. Yet, I find the last sentence of the Conclusions to be a bit “hyped”. I disagree this is an important step for automation in the disciplines, since all step 3 is doing is learning what the geologists manually re-labelled in step 2… How can this be considered automation? Automation was only attempted at step 1 and it failed. Furthermore, it could be argued that determination of clusters by dbscan is never a true automation because of the problematics of generalizing MinPoints and eps (manually selected here as well), which are always dataset-specific and therefore site-specific. I think automation should not be addressed in the paper, or if addressed at least in a more realistic perspective considering what was achieved.

Additional comments

Good job, I believe that with the right adjustments to the manuscript, this contribution might help other palaeontologists working in sites with similar problems of complex/unclear horizon diagnosis. I hope my comments can help you improve the quality of the manuscript.
Below, a list of some details that can easily be improved:
In the abstract “stratigraphical horizons” reads better as “stratigraphic horizons”.
In line 119, the reference protocol cited (Eberth et al, 2007) seems to be missing in References.
In lines 127 and 129 “For the purpose of this study” is repeated.
In line 241 “Considering the amount of data produced by the DBSCAN algorithm” is not clear, i.e. makes it sound dbscan is a simulator/data-generation approach instead of a clustering algorithm. Please rephrase.
Line 351 “exponentially small” is a technical term for describing decay or a tendency in a function.
I highly disagree of using “machine teaching unsupervised algorithms” (lines 443-444), which is not a thing. Maybe you meant “learning” instead of “teaching”? If you are manually adding labels, deleting labels, reordering labels, reclassifying labels, grouping labels, etc. it stops being unsupervised. In fact, “Machine teaching” and “unsupervised algorithms” are almost perfect antonyms!
Best regards,

---

## Round 0.2 · Minor Revisions

Thank you for submitting your revised manuscript. The revisions seem to have enhanced the article. I have only one remaining concern and this relates to an issue flagged by referee #1 on the original manuscript: the provision of the confusion matrix. This is an issue on which I agree with referee #1, indeed in my own community the provision of the confusion matrix is seen a good practice although sometimes it is impractical to show all matrices. I do not wish to push my own views (e.g. the style of presentation of the matrix or the unsuitability of metrics such as kappa derived from it) but do think the core thrust of referee #1's comment is valid and the paper would gain from the provision of the confusion matrix. This would help others interpret the results more easily. Could the matrix be added? I think this would strengthen the article, aid communication and complete the paper.

---

## Round 0.3 · accepted · Accept

Thank you for addressing the issue connected to the confusion matrices; I think including the matrices in the supplementary material is sensible. Thank you for your interesting paper.